# Exploring the association between social determinants and aphasia impairment: A retrospective data integration approach

Molly Jacobs[1], Elizabeth Evans[2], Charles Ellis[2]*

1 Department of Health Services Research, Management and Policy, College of Public Health and Health Professions, University of Florida, Gainesville, FL, United States of America, 2 Department of Speech, Language and Hearing Sciences, Communication Equity and Outcomes Laboratory, College of Public Health and Health Professions, University of Florida, Gainesville, FL, United States of America

* ellisch@phhp.ufl.edu

**Data Availability Statement:** Data from AphasiaBank (https://aphasia.talkbank.org/) and Medical Expenditure Panel Survey (MEPs) (https://meps.ahrq.gov/data_stats/data_use.jsp) were

## Abstract

### Introduction

Traditionally, the study of aphasia focused on brain trauma, clinical biomarkers, and cognitive processes, rarely considering the social determinants of health. This study evaluates the relationship between aphasia impairment and demographic, socioeconomic, and contextual determinants among people with aphasia (PWA).

### Methods

PWA indexed within AphasiaBank—a database populated by multiple clinical aphasiology centers with standardized protocols characterizing language, neuropsychological functioning, and demographic information—were matched with respondents in the Medical Expenditure Panel Survey based on response year, age, sex, race, ethnicity, time post stroke, and mental health status. Generalized log-linear regression models with bootstrapped standard errors evaluated the association between scores on the Western Aphasia Battery-Revised Aphasia Quotient (WAB-R AQ) and demographic, economic, and contextual characteristics accounting for clustering of respondents and the stratification of data collection. Region, age, and income specific models tested the sensitivity of results.

### Results

PWA over age 60 had 2.4% (SE = 0.020) lower WAB-R AQ scores compared with younger PWA. Compared to White PWA, Black and Hispanic PWA had 4.7% (SE = 0.03) and 0.81% (SE = 0.06) lower WAB-R AQ scores, respectively, as did those and living in the Southern US (-2.2%, SE = 0.03) even after controlling for age, family size, and aphasia type. Those living in larger families (β = 0.005, SE = 0.008), with income over $30,000 (β = 0.017, SE = 0.022), and a college degree (β = 0.030, SE = 0.035) had higher WAB-R AQ relative to their counterparts. Region-specific models showed that racial differences were only significant in the South and Midwest, while ethnic differences are only significant in the West. Sex

utilized in this study. Data from MEPS is publicly available and can be obtained online. Utilization of MEPS is governed by a proprietary data use agreement with the Agency for Healthcare Research and Quality (AHRQ) and /or the National Center for Health Statistics (NCHS) and may not be used for any purpose other than for the purpose for which it was supplied as outlined in the data use agreement. Data from AphasiaBank is third-party, but the authors do not have the authority to share these data. Data use privileges were granted to the corresponding author who is a member of the AphasiaBank consortium. If anyone is interested in becoming a member of the AphasiaBank consortium, the information is provided online. These instructions indicate that, "Researchers, educators, and clinicians working with aphasia who are interested in joining the consortium should read the Ground Rules and then send email to macw@cmu.edu with contact information and affiliation, and a brief general statement about how you envision using the data.

**Funding:** The author(s) received no specific funding for this work.

**Competing interests:** The authors have declared that no competing interests exist.

differences only appeared in age-specific models. Racial and ethnic differences were not significant in the high-income group regression.

## Conclusion

These findings support evidence that circumstances in which individuals live, work, and age are significantly associated with their health outcomes including aphasia impairment.

## Introduction

Racial-ethnic differences have been reported consistently in stroke-related outcomes [1–6]. Studies show that Blacks, and to some extent Hispanics, experience higher levels of disability compared with Whites [7] which has largely been attributed to their increased stroke severity and younger age at stroke [2]. Overwhelming evidence links the greater post-stroke impairment and reduced post-stroke functionality among Blacks [2, 8] to multiple social determinants of health (SDOH) such as health literacy, healthcare access, income, education, and residential location [9–11]. Social determinants are influencers or predictors that encompass the conditions in which people live, learn, work, and age. Social determinants can impact health directly but also can indirectly impact health by shaping how people behave within their broader social position [12]. Poverty, unemployment, and housing insecurity are all examples of social determinants that result in poor health outcomes [12, 13]. Social determinants not only concern context or environment, but also include race and socioeconomic status. These determinants are associated with factors such as health behaviors, access to healthcare, and the exposure to contextual toxins and pollutants.

Despite this evidence, few studies have tested the association between these SDOH and deficits among persons with aphasia (PWA)—a poststroke condition characterized by deficits in listening comprehension, oral expression, reading, and writing that can result in significant communication limitations even in its mildest form [14]. Race/ethnicity, gender, socioeconomic status and education have all been associated with post-stroke aphasia impairment. For example, Black PWA demonstrate worse word fluency and auditory comprehension compared to their White counterparts [4]. Additionally, race appears to mediate the relationship between lesion size and aphasia impairment at larger lesion sizes [15]. Furthermore, income and education have been associated with acute aphasia impairment along with life participation in PWA one-year post-onset [16]. However, a recent review by O'Halloran et al., [17] states that there continue to be profound gaps in the examination of the association of the SDOH on language performance in PWA.

In previous work we have suggested that any exploration of disparities in aphasia outcomes must carefully explore not only the SDOH but also studies designed with an intersectional lens of inquiry [18]. Intersectional approaches are utilized to explain how differing lived experiences translate into differences in clinical outcomes [19]. Intersectional approaches begin with careful consideration of childhood and how the early lived experience translate synergistically over to time and the influence of any geographic inequities that impact health over during the lifespan [20]. Most previous work in aphasia examining the association between social determinants and aphasia impairment have not evaluated the SDOH through an intersectional framework despite evidence that the SDOH often work synergistically to impact health outcomes [18].

## Study perspective and approach

This study utilizes a health services research (HSR) approach. HSR explores the intersection of health practice patterns, patient outcomes, and health policy [21] using methods such as regression decomposition, quasi-experimental design, and extended dynamic modelling to determine the relative contributions of factors such as SDOH to clinical health outcomes [22, 23]. Although the SDOH have emerged as key predictors of general health outcomes, less attention has been given their role in rehabilitation and recovery of conditions such as aphasia. To explore this issue in aphasia, this study utilized a HSR approach by integrating clinical data from a national repository and national survey data to replicate the impact of SDOH on aphasia outcomes. Recently, the emergence of analytic approaches has enabled researchers to integrate data related to social determinants with individual clinical data from different sources allowing the systematic examination of SDOH and clinical outcomes [24, 25]. Utilizing these novel data integration and analytic methods, researchers have gained new insights into the impact of SDOH on clinical outcomes [26].

Our recent work utilized a similar approach that involved merging two datasets; the Moss Aphasia Psycholinguistic Project Database (MAPPD) and the 2009–2011 Medical Expenditure Panel Survey (MEPS) to explore the influence of SDOH on naming among individuals with aphasia [27]. The study showed an association between multiple social determinants and naming performance in PWAs. Larger family size and higher income were associated with improved naming performance while race (Black PWA) and low educational attainment were associated with worse naming performance. The study highlighted how the SDOH can magnify the outcome of post-stroke impairments in some population groups. However, despite these findings, the intersectional association between social, economic, and contextual characteristics and disparities in overall post-stroke aphasia language performance is still unclear. Therefore, the objective of this study was to use a large population of PWA to examine the association between aphasia impairment (WAB-R AQ) and SDOH (health, socioeconomic, and contextual) in a diverse cohort of PWA. Generalized linear regression models assessed the association between WAB-R AQ scores and age, income, housing, sex, race, ethnicity, marital status, family size, healthcare utilization, transportation, region of residence, and aphasia type.

## Materials and methods

This retrospective study utilized previously collected data. This study was reviewed and approved by the institutional review board (IRB) as exempt based on the use of deidentified data and the research poses no more than minimal risk. The methodology of this project was previously reported in [27]. *Data Sources*: Two primary data sources were used for this study —1) AphasiaBank [28] and the 2) Medical Expenditure Panel Survey (MEPS) [29].

1. *AphasiaBank*: AphasiaBank is a shared database of individuals with aphasia as well as non-aphasic controls. The database was established in 2005 and is freely available to aphasia researchers and clinicians for educational, clinical, and scholarly uses. AphasiaBank was originally funded by the National Institute on Deafness and Other Communication Disorders (NIDCD) in 2007 and is currently supported by NIDCD grant R01-DC008524 for 2022–2027. The following tests are administered to aphasic participants: Western Aphasia Battery-Revised [30]; Boston Naming Test-Second Edition [31]; Northwestern Assessment of Verbs and Sentences-Revised [32]; and the AphasiaBank Repetition test.

The database contains information on individuals from PWA within and outside of the United States (US) with a variety of primary languages, this analysis includes only those who were tested and treated in the US in the English language. As of December 15, 2023, data from 361

PWA were available in the database. Only individuals with an aphasia etiology identified as stroke, a valid WAB-R score, and relevant demographic data (age, sex, race/ethnicity), and mental health status (presence/absence of depression) were included in the analysis. One hundred and forty-seven individuals with aphasia tested between 2004 and 2020 and with all relevant data were included in the analysis reported here. The average age of the AphasiaBank sample was 60.9 years (SD = 11.74) with an average of 5.41 years (SD = 5.02) post stroke onset. Over half (60%) of the sample was male with 13% Black and 4% Hispanic. In addition to race and ethnicity, aphasia type, age, depressive symptoms, time post stroke onset, education, and WAB-AQ score were also utilized.

2. *MEPS*: The MEPS is a set of large-scale surveys of families and individuals, their medical providers (doctors, hospitals, pharmacies, etc.), and employers across the United States. MEPS collects data on the specific health services and the frequency, cost, and payment for these services. PWA were identified in MEPs using the following criteria: 1) reported they had been previously diagnosed with a stroke by a medical provider, 2) had an ICD-9 (784, 438) or ICD-10 (R43) code for aphasia, and 3) had relevant demographic and mental health status data. The analysis was limited to those MEPS respondents in 2004 through 2020—the years of testing indicated in AphasiaBank—to ensure a robust pool of respondents were available for analysis. The process yielded a total of 693 PWA.

## Integration of datasets

As previously described in [27], an integrated dataset was developed using data elements common to both data sources. AphasiaBank entries were matched with up to three MEPS respondents using a propensity score algorithm [33]. Selection of relevant characteristics of similarity that were uncorrelated with the aphasia impairment score but were present in both databases were identified. Selection of the variables on which to integrate these databases followed the criterion established by Gertler et al., [34] which outlined that subjects should be matched on characteristics 1) that result in a resemblance between population rather than individual-level characteristics and 2) based on expectation of similar circumstance and/or situational context between groups. Since it is impossible to remove all sources of heterogeneity, the statistical process was designed to pair subjects based on key observed covariates [35]. Valid criterion for matching therefore should be observable in both databases, constitute relevant characteristics of similarity, and not be correlated with the ultimate outcome of interest, a.k.a. aphasia impairment score [36].

For this study, age, sex, year of testing/response, race/ethnicity, and mental health status met the criterion for matching between the two datasets of PWA. Respondents with aphasia from AphasiaBank were matched with up to three MEPS respondents based on the identified characteristics of similarity. Since not every AphasiaBank entry has the same number of matches in MEPS, the algorithm is also used to generate a custom set of inverse probability treatment weights that can be used in subsequent analyses to estimate the average treatment effect. In each matched set, each AphasiaBank entry has a weight of 1 and each MEPS respondent has a weight that is computed from contributions of its matched treated units. That is, if an AphasiaBank entry has three matched MEPS respondents, then each MEPS respondent has a weight of 1/3 from this AphasiaBank entry. The total weight for the controls is equal to the total number of MEPS respondents in each matched group, and the total weight for the matched controls is equal to the total number of matched treated units. This is a commonly used approach when evaluating SDOH since most datasets rarely contain all potential determinants of the observed outcomes [37]. Additionally, the approach creates a dataset with greater

**Table 1. Standardized mean differences (AphasiaBank-MEPS).**

|  | Observations | Mean Difference | Std Dev | Standardized Difference | Variance Ratio |
|---|---|---|---|---|---|
| AGE | All | -2.19 | 13.40 | -0.16 | 0.65 |
|  | Matched | -2.33 |  | -0.17 | 0.73 |
| BLACK | All | 0.59 | 0.40 | 1.49 | 0.55 |
|  | Matched | 0.42 |  | 1.07 | 0.45 |
| FEMALE | All | 0.22 | 0.49 | 0.45 | 1.05 |
|  | Matched | 0.15 |  | 0.31 | 0.99 |
| WHITE | All | -0.17 | 0.43 | -0.39 | 0.65 |
|  | Matched | -0.13 |  | -0.29 | 0.69 |
| HISPANIC | All | 0.13 | 0.29 | 0.46 | 0.21 |
|  | Matched | 0.05 |  | 0.18 | 0.38 |
| DEPRESSED | All | 0.03 | 0.37 | 0.08 | 0.86 |
|  | Matched | 0.00 |  | 0.01 | 0.97 |

Standard deviation of all observations used to compute standardized differences

statistical power to explore the outcomes of interest than would be obtained by using either one individually [38]. Respondents were matched with replacement to preserve the distributional integrity of the data. The resulting data matched AphasiaBank (N = 147) entries with up to three members of the MEPS post-stroke aphasia cohort (N = 693) resulting in an integrated data set (n = 402). Basic demographic characteristics of the integrated data set is listed in Table 1.

## Specification of covariates

The integrated data contained all elements necessary to estimate the relationship between aphasia impairment and individual, socioeconomic, and contextual determinants. Individual characteristics included in the regression model were age ($\leq$60,* >60), sex (Female, Male*), race (Black, White*), ethnicity (Hispanic, Non-Hispanic*), region of residence (Northeast,* North Central/Midwest, South, West), family size (1 to 9), income (<$30,000, $\geq$$30,000*), insurance status (insured,* uninsured), education (less than college graduation, college graduate or above*), time post aphasia onset, and aphasia type (Anomic,* Broca's, Global). Since literature outlining social determinants of aphasia is sparse, these covariates were selected based on a review of recently published studies outlining the determinants of stroke [11, 39–41].

## Dependent variable

The Western Aphasia Battery-Revised (WAB-R) [30] was used to classify aphasia by classical type, measure overall severity, and measure change over time. The WAB-R is a commonly used diagnostic tool used to assess the linguistic skills and main nonlinguistic skills of adults with aphasia. The WAB-R includes eight individual subtests the collectively provide information for the diagnosis of the type of aphasia and identifies the location of the lesion causing aphasia. The WAB-R subtest scores yield an Aphasia Quotient (AQ) which is a weighted average of all subtest scores relating to spoken language, measuring language ability that renders scores on a scale from one to 100.

## Empirical model

The relationship between individual WAB-R-AQ score and demographic, economic, and contextual determinants was estimated using a generalized log-linear estimation equation

regression model. To account for the multistage probability sampling used by the MEPS, models specify primary sampling units to account for clustering within housing units and geographic strata to indicate relative densities of population characteristics. Differences in the scales and numeric ranges of model factors such as age, WAB-AQ, family size, and time post onset can yield uninterpretable regression coefficients. Therefore, logarithmic transformations of these continuous variables were used to scale these parameters. Scaling has the advantage of reducing the potential for multicollinearity in a regression model while preserving the distribution of the scaled parameters. Use of these transformed values resulted in regression coefficients that could be interpreted as relative percentage changes.

To observe how the of additional covariates change the model estimates, we began by including only demographic characteristics (age, sex, race, and ethnicity in the model) then added family and community characteristics (family size and region of residence). Next, income and resource related characteristics were included (insurance and income) and finally covariates indicating aphasia type and time post onset were included. This iterative inclusion process ensured efficient and accurate estimation. Regression weights were used to account for differences in the number of matched pairs and bootstrapping was performed to correct estimate standard errors and ensure the robustness of findings. To facilitate the interpretation of findings, average marginal effects are calculated for each independent variable.

**Sensitivity analysis.** To test the sensitivity of results to age, regional, and income differentials, additional analyses were performed. These analyses assessed each region (Northeast, North Central/Midwest, South, West), age group ($\leq$60,* >60), and income segment (<\$30,000, $\geq$\$30,000*) separately.

## Results

Table 1 shows the differences between the integrated and unintegrated data sets by demographic characteristics. Differences were nominal and were within an expected range. After integration, a sample of 402 PWA was used for the analysis. Table 2 provide the demographic characteristics of the integrated sample used in the analysis and statistically differences between population subgroups, respectively. Mean WAB-AQ score was 69.25 (SD = 19.28), average time post onset was 5.60 (SD = 5.12), and the average family size was 2.09 (SD = 1.53) with no significant differences across the sample groups (WAB-AQ: F = 0.14, p = 0.7104; TPO: F = 0.08, p = 0.7719; family size: F = 0.53, p = 0.4669).

About 47% of the sample was age 60 or below with 53% being older. Black PWA were slightly younger with 82.35% below age 60 compared to 39% and 25% of White and Hispanic PWA, respectively ($\chi^2$ = 28.6, p<0.0001). The full sample was 58.29% male and 41.71%, but subgroups varied significantly ($\chi^2$ = 14.6, p < .0001). Whites were 60.91% male compared to 47.06% and 50% of Blacks and Hispanics. One-third of White PWA had income above \$30,000, compared to 13.73% of Blacks and 16.67% Hispanics—a statistically significant difference ($\chi^2$ = 63.7, p<0.0001).

The sample was distributed across the US with 39.59% residing in the South, 24.53% in the Midwest, 20.9% in the West, and 14.97% in the Northeast, but these proportions varied across subgroups ($\chi^2$ = 46.6, p < .0001). Only 8.67% of the sample had a college degree with the highest proportion of college graduates being White (9.51%), 5.30% Black, and 0% Hispanic ($\chi^2$ = 19.6, p<0.0001). Most individuals had anomic aphasia (49.46%) or Broca's (47.83%) aphasia with only 2.71% having global aphasia.

Table 3 contains estimates of the generalized linear regression. Each additional year post onset was associated with a 4.5% (SE = 1.2%) increase in WAB-AQ score. Similarly, each additional member of the family in the household increased WAB-AQ by 0.5% (SE = 0.8%). PWA

**Table 2. Sample means and frequencies with test for subgroup differences.**

|  | Sample (N = 402, 100%) | | White (N = 332, 82.59%) | | Black (N = 54, 13.43%) | | Hispanic (N = 16, 3.98%) | | | |
|---|---|---|---|---|---|---|---|---|---|---|
|  | Mean | Std Dev | Mean | Std Dev | Mean | Std Dev | Mean | Std Dev | F-Statistic | p-Value |
| **WAB-AQ (20.2–93.2)** | **69.25** | **19.28** | **70.04** | **19.43** | **65.96** | **19.73** | **69** | **14.58** | **0.14** | **0.7104** |
| **TPO (1–32)** | **5.6** | **5.12** | **5.53** | **5.2** | **5.55** | **5.27** | **6.15** | **2.34** | **0.08** | **0.7719** |
| **Family Size (1–9)** | **2.09** | **1.53** | **2.08** | **1.53** | **1.92** | **1.25** | **2.62** | **2.43** | **0.53** | **0.4669** |
|  | N | Percent | N | Percent | N | Percent | N | Percent | $\chi^2$ | p-Value |
| Age > 60 | 219 | 53.32 | 202 | 60.91 | 11 | 17.65 | 13 | 75 | 28.6 | < .0001 |
| Income ≥ $30,000 | 100 | 30.2 | 91 | 27.27 | 10 | 13.73 | 4 | 16.67 | 63.7 | < .0001 |
| Female | 168 | 41.71 | 130 | 39.09 | 30 | 52.94 | 8 | 50 | 14.6 | < .0001 |
| Uninsured | 31 | 4.57 | 26 | 7.58 | 5 | 5.88 |  |  | 30.2 | < .0001 |
| College Degree | 27 | 8.67 | 24 | 6.97 | 5 | 5.88 |  | 100 | 19.6 | < .0001 |
| Global Aphasia | 11 | 2.71 | 7 | 1.82 | 4 | 5.88 |  | 30 |  | < .0001 |
| Broca's Aphasia | 193 | 48.04 | 155 | 47.27 | 25 | 47.06 | 8 | 50 |  |  |
| Midwest | 107 | 24.53 | 89 | 26.67 | 16 | 29.41 | 4 | 16.67 | 46.6 | < .0001 |
| South | 155 | 39.59 | 129 | 38.79 | 19 | 35.29 | 7 | 50 |  |  |
| West | 90 | 20.9 | 77 | 23.33 | 9 | 17.65 | 3 | 25 |  |  |

Estimates weighted to reflect nationally representative population and adjust for matched sample

**Table 3. Social, economic, and contextual associations with aphasia impairment.**

| **R-Square** | **0.7321** | | | |
|---|---|---|---|---|
| **Adj R-Square** | **0.722** | | | |
| **F-Value** | **72.23** | **< .0001** | | |
|  | Estimate | Std Error | t Value | Pr > \|t\| |
| Intercept | **4.357** | 0.039 | 111.390 | < .0001 |
| Age > 60 | **-0.024** | 0.020 | -2.210 | 0.023 |
| Income > $30,000 | **0.017** | 0.022 | 2.790 | 0.043 |
| TPO | **0.045** | 0.012 | 3.780 | 0.000 |
| Family Size | **0.005** | 0.008 | 2.660 | 0.051 |
| Female | 0.032 | 0.019 | 1.680 | 0.093 |
| Black | **-0.047** | 0.030 | -3.570 | 0.012 |
| Hispanic | **-0.008** | 0.056 | -2.130 | 0.019 |
| Uninsured | -0.037 | 0.046 | -0.800 | 0.422 |
| College Degree | **0.030** | 0.035 | 2.850 | 0.040 |
| Global Aphasia | **-1.316** | 0.060 | -21.770 | < .0001 |
| Broca's Aphasia | **-0.477** | 0.019 | -24.490 | < .0001 |
| Midwest | -0.004 | 0.032 | -0.130 | 0.899 |
| South | **-0.022** | 0.029 | -2.750 | 0.045 |
| West | 0.010 | 0.032 | 0.300 | 0.761 |

Dependent Variable: WAB-AQ

Estimates weighted to reflect nationally representative population and adjust for matched sample

*Indicates* significant at 95% confidence level.

Reference Category: Age (≤60), Income (low income ≤$30,000+), Insurance (insured), Race (White), Ethnicity (Non-Hispanic), Sex (Male), Aphasia Type (Anomic), Education (Less than a college degree), Region (Northeast)

with income above $30,000 had 1.7% high WAB-AQ than those with lower income, while those with a college degree at 3% (SE = 3.5%) higher score than those without. However, relative to Whites, Black and Hispanic PWA had 4.7% (SE = 3.0%) and 0.8% (SE = 5.6%) lower WAB-AQ. Compared to residents of the Northeast, PWA residing in the South had 2.2% (2.9%) lower scores, all else held constant. As expected, individuals with Broca's ($\beta$ = -0.477, SE = 0.019) and global ($\beta$ = -1.316, SE = 0.060) aphasia had lower WAB-AQ than those with anomic aphasia.

**Region, age, and income group analyses.**  Tables A1-A3 in S1 Appendix provide the region, age, and income groups models respectively. As expected, coefficient magnitudes varied between subgroups and significance patterns differ slightly. In the regional models (Table A1 in S1 Appendix), racial differences were only statistically significant in the South and Midwest, while ethnic differences only appeared in the West. The age-group model (Table A2 in S1 Appendix) showed that, among those aged 60 and under, Blacks had 3.1% (SE = 3.8%) lower WAB-AQ compared to Whites, but scores were not statistically different among those over 60. However, female over 60 had 9.20% higher WAB-AQ compared to males over 60—a difference not observed among the younger age group.

Finally, income group models (Table A3 in S1 Appendix) showed 2.5% (SE = 0.033) and 0.8% (SE = 6.0%) lower WAB-AQ among Blacks and Hispanics earning less than $30,000, but no racial or ethnic differences among the higher income group. However, higher income earners did show a negative relationship between family size ($\beta$ = -0.047, SE = 0.018) and WAB-AQ score.

## Discussion

The objective of this study was to examine the association between SDOHs and aphasia outcomes using a novel, integrated database. In this retrospective analysis, the primary research was "Do specific SDOH influence aphasia outcomes". Findings suggested that several SDOH were significantly associated with aphasia outcomes.

There was an association between age, time post-onset, and race/ethnicity on language performance. Additionally, family size and access to resources such as higher income were associated with higher language performance scores in PWA. Lastly, being uninsured and those with a condition-related care visit in a college degree were associated with higher language performance. However, two main differences in associations were revealed when subgroups of region, age and income groups were further examined. There was a positive association between age and WAB-R AQ scores in the Northeast, Midwest, and West, but a slightly negative association in the South. Additionally, differentials between race and ethnicity and WAB-R AQ performance significantly increased with income.

### Socio-demographic factors

**Race/ethnicity.**  Regarding race/ethnicity specifically, recent work consistently demonstrates racial ethnic differences in stroke outcomes. Blacks and Hispanics demonstrate higher aphasia impairment along with higher healthcare expenditures [4, 42, 43]. The relationship between level of impairment and healthcare expenditures is straightforward as individuals with greater stroke-related disabilities requiring greater levels of care have longer lengths of stay, greater utilization and subsequently higher costs of care [44]. Similarly, individuals with greater post-stroke impairments such as aphasia also have greater costs of care [45]. However, the underlying cause of greater disability following stroke and post-stroke impairment in conditions such as aphasia among Blacks is less understood.

A number of factors have been proposed to explain racial differences in aphasia outcomes such as differences in the underlying cause of stroke, differences in the post-stroke recovery environment, therapies received and access to rehabilitation care [4]. Contemporary models argue that earlier onset of comorbid conditions such as diabetes and hypertension [46–48] along with the weathering process among racial-ethnic minorities [49] contribute to differential pathways of aging leading up the stroke and subsequently post-stroke outcomes [50]. These concepts have not been tied specifically to poststroke conditions such as aphasia however they offer opportunities for novel exploration.

**Age.** Regarding age and aphasia outcomes, a review by Ellis et al., [51] concluded that younger patients with aphasia were more likely to exhibit non-fluent or Broca's type of aphasia and studies examining aphasia recovery and aphasia clinical outcomes did not demonstrate a direct relationship between age and aphasia recovery. Consequently, age only impacted likelihood of aphasia and aphasia type. If this assumption is correct, the observation of lower aphasia impairment in older adults may reflect other relationships with older age. Older adults frequently have Medicare which may offer greater access to rehabilitation [52]. For example, Medford-Davis et al., [52] found that the privately insured were less likely to die in the hospital and more likely to go to inpatient rehab. Similarly, patients with Medicare were more likely to receive rehabilitation during hospitalization and receive transfer to a rehabilitation facility. It is notable that these findings are specific to general stroke rehabilitation care, and it is unclear how these observations translate to better rehabilitation of post-stroke conditions such as aphasia. Finally, it is possible that aphasia impairment among older adults is less severe due to less severe strokes.

**Education.** Regarding education, there is not a consensus for the relationship of education with aphasia outcomes. Worrall et al., [16] found more education to be associated with reduced life participation while González-Fernández et al., [53] found more education to be associated with improved language performance in persons with aphasia. However, O'Halloran et al., [17] did not identify sufficient evidence that education level influences aphasia outcomes. A possible explanation is that more education improves language performance on more formal assessments but is not as protective for health-related quality-of-life factors.

**Income and resources.** Our finding that higher income aligns with previous findings demonstrating higher income improves stroke outcomes [54, 55]. Access to wealth even beyond income and presence/absence of insurance translates into better outcomes [56]. Higher incomes are linked to lower rates of disease, less premature death and generally better health and higher quality of care [56, 57]. The relationship between income and health is gradient or stepwise in improvements at different economic lives [56]. Whether measured from the perspective of the individual or healthcare systems, availability of resources that are linked to wealth matter in relationship to optimal health outcomes. Healthcare systems with greater resources can provide higher quality care which translates into improved outcomes among patients [58]. Further, improved quality of care must consider resources from multiple perspectives because healthcare systems are dynamic and are the function of a) the inter-relationships between the patient, clinical and nonclinical healthcare workers, b) the differing levels of the health system (community to tertiary referrals and c) required human and material resources. It may be that more resources, receipt of better quality of care, and the progression of recovery can develop naturally and synergistically thereby improving outcomes. Further study will be required to adequately determine how this translates to post-stroke conditions such as aphasia. Lastly, being uninsured was associated with higher language performance. This contradicts previous findings where having insurance improved stroke outcomes particularly in the outpatient setting where substantial rehabilitation occurs [59]. It is tenable that many of the uninsured were younger (under at 65 thus not covered by Medicare). Younger

adults are more resilient in many ways and more likely to recover faster or more completely [60]. Additionally, in this work we examined "condition-specific visits" and in this case related to aphasia. Those receiving the most condition-specific visits also had the best aphasia recovery scores. More specifically the condition-related visit means that they received care from a healthcare provider for their aphasia (i.e. speech language pathologist-SLP) resulting in improved aphasia outcomes. In this analysis we found the uninsured received more condition-related visits.

## Family and community characteristics

**Family size.** In this study a larger family size was associated with improved language performance. Previous literature has found social support to be associated with acute aphasia impairment and life participation. We found similar findings in our prior work exploring aphasia naming ability which showed a strong association between family size and naming performance [27]. The general stroke literature has shown that families play a key role in stroke recovery [61]. The relationship between families and aphasia recovery is less clear yet families engaging in the lives of stroke survivors has been shown to be related to better recovery [62]. Similarly, the aphasia literature has highlighted the contribution that families offer to the social networks of individuals with aphasia and specifically quality of life [63] via reductions in social isolation [64]. However, we believe family and community offer intersection advantages and disadvantages which carefully considered within the context of the stroke survivors lived environment. Skolarus et al., [65] argue that recovery occurs in three phases: acute stroke period, early recovery period and community living period and aspect of disparities emerge during the final period. It is during this period that differential community resources and access translate into differences in functional capacity to perform pre-stroke activities. Communication which is impacted by aphasia would be differentially impacted by the number of individuals engaged in the communication process in the home and community. Furthermore, the relationship between social support and aphasia outcomes may differ among racial ethnic groups and/or outcome measured. While social support has been associated with life participation outcomes in persons with aphasia [16] social support and social network size in a population of specifically African American persons with aphasia did not predict health-related quality of life [66].

## Intersectional analyses

**Region, age, income.** Highlighting the importance of the intersectional contribution of the SDOH to aphasia outcomes are the findings related to the US region of residence, age and income specific analyses. The reason for the observed association between age and WAB-R AQ is not clear. Although we controlled for time post onset, is possible that the relationship of older age and high WAB-R AQ scores reflect older individuals with aphasia having longer time to recover. Even though we control for TPO, it is highly likely that individuals at older ages have longer time post onset. Also, individuals at older ages are likely on Medicare which covers SLP services and other acute care for stroke. These coverages could explain some of the association. Also, older adults may be retired and able to devote more time specifically to the recovery process rather than other life issues that occur at younger ages. Finally, the larger disparities between racial groups at higher income levels suggest that income does not surmount the vast array of other social determinants that impact minority races and in fact widens the gap in access. Income earned today cannot compensate for the confounding influence of SDOH. All low-income individuals face hardships, so differentials are less pronounced. However, in the upper income range, the multiplicative influence of social, political, and structural inequities on vulnerable populations can been seen more clearly.

### Study limitations

Despite the interesting findings reported here, the study has several limitations. First, this study integrated two different datasets—AphasiaBank and MEPS—which were not originally designed for this type of versatility. In addition, in this kind of work it impossible to account for all of the potential heterogeneity and unobserved endogeniety in the two data sources. Second, the study was completed retrospectively which does not offer the same level of evidence as data collected in a prospective fashion. For example, the AphasiaBank repository includes data collected from experimental research studies over a long period of time. Additionally, many records did not contain the necessary data for this type of analysis. Ideally, future studies will be prospective, and all specific data related to aphasia and SDOH would be collected within one planned study.

Third, this study does not account for clinical variables known to influence to influence post-stroke outcomes such as stroke severity, and comorbid disease conditions that were not available in the databases utilized in this study. Fourth, other variables measuring access to rehabilitation care such as distance from quality healthcare were also not available but must be considered in explanations of the recovery process. Fifth, the psychometric properties of the WAB-R AQ have previous faced criticism [67–69]. However, the WAB-R remains the most widely accepted and utilized clinical outcome measure of aphasia impairment and most widely used measure in studies of aphasia research outcomes [70]. Sixth, diagnosis of stroke was based on self-report which can be limited by the possibility of providing invalid answers.

### Conclusion

This study supports previous literature emphasizing the need for examination of the influence of a wide variety of the SDOH on aphasia outcomes and specifically language performance. Furthermore, intersectional analysis demonstrates the compounding effect of social, economic, and contextual factors on aphasia outcomes. More importantly, future research is necessary to continue identifying the SDOH most associated with aphasia outcomes.

### Supporting information

**S1 Appendix.**
(DOCX)

### Author Contributions

**Conceptualization:** Molly Jacobs, Elizabeth Evans, Charles Ellis.

**Data curation:** Molly Jacobs.

**Formal analysis:** Molly Jacobs.

**Investigation:** Charles Ellis.

**Methodology:** Molly Jacobs, Charles Ellis.

**Writing – original draft:** Molly Jacobs, Elizabeth Evans, Charles Ellis.

**Writing – review & editing:** Molly Jacobs, Elizabeth Evans, Charles Ellis.

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
