## [Decision Letter · Decision Letter 0]

14 Jan 2024

PONE-D-23-42747Exploring the Association Between Social Determinants and Aphasia Impairment: A Retrospective Data Integration ApproachPLOS ONE

Dear Dr. Ellis,

Thank you for submitting your manuscript to PLOS ONE. After careful consideration, we feel that it has merit but does not fully meet PLOS ONE’s publication criteria as it currently stands. Therefore, we invite you to submit a revised version of the manuscript that addresses the points raised during the review process.

We look forward to receiving your revised manuscript.

Kind regards,

Yuyan Wang, Ph.D.

Academic Editor

PLOS ONE

Journal Requirements:

Additional Editor Comments:

This paper is skillfully crafted, utilizing data from AphasiaBank in conjunction with respondents from the Medical Expenditure Panel Survey to explore the relationship between aphasia impairment and various demographic, socioeconomic, and contextual factors in individuals with aphasia. The manuscript is well-written, employing appropriate analytical methods, and the discussions are concise and focused. There are only a few minor comments from the two reviewers. Please kindly address these comments and revise the manuscript accordingly.

Reviewers' comments:

Reviewer's Responses to Questions

**Comments to the Author**

1. Is the manuscript technically sound, and do the data support the conclusions?

Reviewer #1: Yes

Reviewer #2: Yes

2. Has the statistical analysis been performed appropriately and rigorously? 

Reviewer #1: Yes

Reviewer #2: Yes

3. Have the authors made all data underlying the findings in their manuscript fully available?

Reviewer #1: Yes

Reviewer #2: No

4. Is the manuscript presented in an intelligible fashion and written in standard English?

Reviewer #1: Yes

Reviewer #2: Yes

5. Review Comments to the Author

Reviewer #1: The authors utilize AphasiaBank data alongside respondents from the Medical Expenditure Panel Survey to examine the connection between aphasia impairment and various demographic, socioeconomic, and contextual factors among individuals with aphasia. The manuscript is skillfully written, employing appropriate analysis methods and the discussions are succinct and targeted. I only have minor comments. Line 97: has enabled researchers to researchers -> has enabled researchers. For table 2, it would be more reader-friendly if it adheres to the standard baseline table format.

Reviewer #2: The authors presented a nicely written report about the association between SDOH factors and the aphasia outcomes. The method used to integrate the data was solid and the statistical analysis of the integrated data was sound and clear. The results were also presented in a clear fashion, with the discussions combining the study results with additional literatures. My only minor comment is for the authors to report effect sizes and the SE’s in the same format, e.g. 2.5% change (SE = 3.3%).

6. PLOS authors have the option to publish the peer review history of their article (what does this mean?). If published, this will include your full peer review and any attached files.

Reviewer #1: No

Reviewer #2: No

---

## [Author Response · Author response to Decision Letter 0]

24 Jan 2024

General Editorial Comments:

We have reviewed and ensure the manuscript meet’s the journal style requirements.

Please review your reference list to ensure that it is complete and correct. If you have cited papers that have been retracted, please include the rationale for doing so in the manuscript text, or remove these references and replace them with relevant current references. 

The reference list has been reviewed as recommended.

The PLOS Data policy requires authors to make all data underlying the findings described in their manuscript fully available without restriction, with rare exception (please refer to the Data Availability Statement in the manuscript PDF file). The data should be provided as part of the manuscript or its supporting information, or deposited to a public repository. 

We have been responsive to the data policy as outlined in the appropriate section.

Response to Reviewers

Reviewer #1: I only have minor comments. Line 97: has enabled researchers to researchers -> has enabled researchers. 

Revised as recommended.

For table 2, it would be more reader-friendly if it adheres to the standard baseline table format.

The table has been revised as recommended.

Reviewer #2: 

My only minor comment is for the authors to report effect sizes and the SE’s in the same format, e.g. 2.5% change (SE = 3.3%).

This change has been made as recommended.

---

## [Decision Letter · Decision Letter 1]

20 Feb 2024

Exploring the Association Between Social Determinants and Aphasia Impairment: A Retrospective Data Integration Approach

PONE-D-23-42747R1

Dear Dr. Ellis,

We’re pleased to inform you that your manuscript has been judged scientifically suitable for publication and will be formally accepted for publication once it meets all outstanding technical requirements.

Kind regards,

Yuyan Wang, Ph.D.

Academic Editor

PLOS ONE

Additional Editor Comments (optional):

Reviewers' comments:

Reviewer's Responses to Questions

**Comments to the Author**

1. If the authors have adequately addressed your comments raised in a previous round of review and you feel that this manuscript is now acceptable for publication, you may indicate that here to bypass the “Comments to the Author” section, enter your conflict of interest statement in the “Confidential to Editor” section, and submit your "Accept" recommendation.

Reviewer #1: All comments have been addressed

Reviewer #2: All comments have been addressed

2. Is the manuscript technically sound, and do the data support the conclusions?

Reviewer #1: Yes

Reviewer #2: (No Response)

3. Has the statistical analysis been performed appropriately and rigorously? 

Reviewer #1: Yes

Reviewer #2: (No Response)

4. Have the authors made all data underlying the findings in their manuscript fully available?

Reviewer #1: Yes

Reviewer #2: (No Response)

5. Is the manuscript presented in an intelligible fashion and written in standard English?

Reviewer #1: Yes

Reviewer #2: (No Response)

6. Review Comments to the Author

Reviewer #1: (No Response)

Reviewer #2: (No Response)

7. PLOS authors have the option to publish the peer review history of their article (what does this mean?). If published, this will include your full peer review and any attached files.

Reviewer #1: No

Reviewer #2: No

---

## [Editor Report · Acceptance letter]

12 Mar 2024

PONE-D-23-42747R1 

PLOS ONE

Dear Dr. Ellis, 

I'm pleased to inform you that your manuscript has been deemed suitable for publication in PLOS ONE. Congratulations! Your manuscript is now being handed over to our production team.

Kind regards, 

on behalf of

Dr. Yuyan Wang 

Academic Editor

PLOS ONE